# Ovary Dissection Is a Sensitive Measure of Sterility in *Anopheles gambiae* Exposed to the Insect Growth Regulator Pyriproxyfen

**DOI:** 10.3390/insects14060552

**Published:** 2023-06-14

**Authors:** Alina Soto, Mark Rowland, Louisa A. Messenger, Mathew Kirby, Franklin W. Mosha, Alphaxard Manjurano, Natacha Protopopoff

**Affiliations:** 1Department of Disease Control, London School of Hygiene & Tropical Medicine, London WC1E 7HT, UK; 2Department of Environmental and Occupational Health, School of Public Health, University of Nevada, Las Vegas, NV 89557, USA; 3PMI VectorLink Project, Abt Associates, 6130 Executive Blvd., Rockville, MD 20852, USA; 4Pan African Malaria Vector Research Consortium, Kilimanjaro Christian Medical University College, Moshi P.O. Box 2240, Tanzania; 5National Institute for Medical Research Tanzania, Mwanza P.O. Box 9653, Tanzania

**Keywords:** pyriproxyfen, insect growth regulator, *Anopheles gambiae*, malaria control, long-lasting insecticidal nets, pyrethroid resistance, fecundity, sterility

## Abstract

**Simple Summary:**

Malaria is transmitted by *Anopheles* mosquitoes. An important control tool against malaria is insecticidal bed nets containing a pyrethroid insecticide and another active ingredient, such as pyriproxyfen (PPF), which sterilizes adult mosquitoes by disrupting egg development. To evaluate the impact of PPF-treated bed nets, there are two potential techniques available to measure mosquito sterility: (1) observing egg-laying rates and (2) ovarial dissection. Our aim was to show which technique was most effective by exposing adult mosquitoes to PPF-treated or untreated nets and comparing sterility outcomes. When identifying the true percentage of mosquitoes sterilized from PPF exposure (i.e., true positives), both techniques showed similar sensitivity (egg-laying: 99% vs. dissection: 100%). For identifying non-exposed mosquitoes in the untreated group (i.e., true negatives), dissection had a higher specificity than egg-laying (53% vs. 19%). To show that ovary dissection remains effective under different conditions, we exposed mosquitoes to insecticides using two different exposure methods, including nets treated with pyrethroids, and blinded the investigator to reduce bias when scoring. The investigator predicted the exposure status of dissected females across all conditions with >90% accuracy. We conclude that ovary dissection is a more sensitive and specific tool for measuring sterility in *Anopheles* mosquitoes exposed to PPF.

**Abstract:**

Pyriproxyfen (PPF) is an insect growth regulator used in the co-treatment of long-lasting insecticidal nets for its ability to sterilize female mosquitoes. To evaluate the efficacy of PPF-treated nets on mosquito reproductivity, most studies observe oviposition (egg-laying) rates in the laboratory. This technique has several technical disadvantages. Our study assessed if ovarial dissection could serve as an effective proxy for evaluating sterility in *Anopheles gambiae* mosquitoes. Blood-fed females were exposed to untreated or PPF-treated nets in cylinder assays and followed over several days to observe oviposition rates or egg development by dissection. For identifying PPF-exposed mosquitoes, both techniques demonstrated high sensitivity (oviposition: 99.1%; dissection: 100.0%), but for identifying non-exposed mosquitoes, specificity was significantly higher in the dissection group (52.5% vs. 18.9%). To assess whether dissection could be applied to nets treated with a pyrethroid or co-treated with a pyrethroid and PPF in tunnel tests, a blinded investigator performed dissections to predict the PPF exposure status across different treatment groups. The exposure status of dissected females was predicted with >90% accuracy. We report that dissection is a sensitive technique to assess sterility in female Anopheles gambiae mosquitoes and can be used as a predictor of PPF exposure.

## 1. Introduction

Malaria is transmitted to humans by the bites of female *Anopheles* mosquitoes. In the reproductive cycle of mosquitoes, the females must blood-feed to obtain the necessary nutrients for egg development. For this reason, long-lasting insecticidal nets (LLINs), which intercept host-seeking mosquitoes, are a successful control tool against malaria. LLINs were responsible for considerable reductions in malaria transmission until their success was threatened by widespread insecticide resistance in *Anopheles* (*An.*) mosquitoes [1]. To counteract pyrethroid resistance, novel vector control tools, particularly LLINs, containing two active ingredients (AIs) have been developed. The insect growth regulator pyriproxyfen (PPF) is a strong candidate for dual-AI LLINs because it is a potent sterilant: three minutes of tarsal exposure to PPF can induce lifelong sterilization in adult female mosquitoes [2]. The principle behind LLINs co-treated with PPF and a pyrethroid in resistance management is that the pyrethroid component may kill or knock down susceptible mosquitoes, while PPF sterilizes any pyrethroid-resistant survivors, preventing these individuals from producing viable offspring [3]. Two commercially manufactured dual-AI LLINS are Olyset^®^ Duo and Royal Guard^®^, impregnated with a combination of PPF and permethrin or alpha-cypermethrin, respectively.

In 2022, the WHO published a guideline to assess the sterilizing effect of pyriproxyfen in the framework of insecticide resistance management [4], but there are no specific WHO guidelines for evaluating the bio-efficacy of LLINs containing growth regulators [5]. Most laboratory studies have evaluated the impact of PPF on adult sterility by measuring egg development (fecundity) and egg hatching (fertility) outcomes [2,6,7,8,9,10,11]. This is typically performed by exposing adult female mosquitoes to PPF before or after a bloodmeal, providing them with an egg-laying (oviposition) substrate, and recording the number of eggs laid and the proportion of emerged larvae, pupae, and F_1_ adults. These measurements provide estimates of reductions in fecundity, fertility, and reproductive rates caused by PPF exposure. Unfortunately, a disadvantage often observed with this method is that wild-caught females do not readily oviposit under the artificial confines of the laboratory, resulting in low oviposition rates and an underestimated impact of PPF exposure [12]. Additionally, this method requires suitable equipment and rearing facilities with potentially long follow-up times [13].

Dissecting gravid females to measure egg development may be an alternate technique to evaluate PPF-induced sterility. Because PPF causes arrested development in eggs, this effect can be observed microscopically. Evidently, studies have shown clear differences in the development of *An. gambiae* eggs after exposure to PPF [2,14]. The aims of our study were: (1) to evaluate if ovarial dissection can be used to measure sterility outcomes (as proxies for oviposition studies) in female *An. gambiae* exposed to PPF and pyrethroid LLINs, and (2) to test whether dissection can be used to predict the exposure status (i.e., PPF-exposed or non-exposed) of female *An. gambiae* mosquitoes.

## 2. Materials and Methods

### 2.1. Mosquitoes

Two pyrethroid-resistant *Anopheles gambiae sensu stricto* (s.s.) strains were used in this study: the RSP strain with the knockdown resistance mutation VGSC-1014S, and the Muleba-Kis strain with established VGSC-1014S and metabolic resistance mediated by overexpression of Cyp6 cytochrome p450 enzymes. Mosquitoes were obtained from the insectary at the Pan African Malaria Vector Research Consortium (PAMVERC) in Moshi, Tanzania.

For blood-feeding mosquitoes, guinea pigs were sedated using xylazine and placed over a mesh cage to allow mosquitoes to probe through and take blood from the underside of the guinea pig. The mosquitoes were allowed to feed for 1 h either the same morning or the evening prior to insecticide exposure in cylinder assays. Mosquitoes were sugar-starved for at least 1 h before blood-feeding.

### 2.2. Test LLINs

The following LLINs obtained from Sumitomo Chemical Co. Ltd., Tokyo, Japan were tested in cylinder assays and tunnel tests:(1)Olyset^®^ Duo (2% permethrin and 1% PPF), unwashed;(2)Olyset^®^ Duo (2% permethrin and 1% PPF) with 20 washes (20W);(3)Olyset^®^ Net (2% permethrin), unwashed;(4)Long-lasting PPF Net (1% PPF), unwashed;(5)Long-lasting PPF Net (1% PPF), 20W.

An Untreated net was included as a control (polyester net, A to Z Textile Mills Ltd., Arusha, Tanzania). Olyset^®^ Duo 20W and PPF Net 20W were washed following WHO guidelines [5].

### 2.3. Bioassay Procedures

Cylinder assays were conducted using the *An. gambiae* s.s. RSP and Muleba-Kis strains. Net pieces (12 cm × 15 cm) were stapled onto a blank piece of paper and inserted into the exposure tubes. A total of 10 to 20 recently blood-fed 2–5 day old females were aspirated into holding tubes and left to acclimatize for 1 h, then transferred to exposure tubes for 3 min. Following exposure, mosquitoes from each replicate were placed into individual cups and provided with 10% glucose solution on cotton pads ad libitum. Mortality was recorded at 24, 48, and 72 h.

Tunnel tests were carried out using the *An. gambiae* s.s. Muleba-Kis strain, following standard WHO guidelines [5]. The tunnels consisted of a square glass cylinder (60 cm × 25 cm × 25 cm) divided into two sections with a framed net piece held between each section. The net piece was deliberately holed for mosquitoes to pass through. A guinea pig was placed behind the net piece and mosquitoes were released into the opposite section. Mosquitoes were able to fly freely toward the guinea pig and cross the net piece for a bloodmeal. Between 30 and 50 unfed 3–6 day old female *An. gambiae* s.s. Muleba-Kis mosquitoes were released into the tunnels and given 14 h of overnight exposure. Live blood-fed females were collected and provided with 10% glucose solution on cotton pads ad libitum. Mortality was recorded at 24 and 72 h.

### 2.4. Oviposition Method

Following LLIN exposure in the cylinder assay, an equal number of individual mosquitoes per treatment group were randomly allocated to paper cups containing either an oviposition substrate (oviposition group) or nothing (dissection group).

Two types of oviposition substrates were tested in the paper cups: (1) a single piece of cotton placed at the bottom of a paper cup was soaked in distilled water and overlaid with a piece of filter paper (for females to oviposit directly on the filter paper); or (2) a piece of filter paper was spread along the bottom edges of the cup and distilled water was added to fill one-third of the cup (for females to oviposit directly on the water). The proportion of ovipositing females did not significantly differ between the two oviposition substrates in terms of the proportion of females that laid eggs (odds ratio; OR: 0.76, *p* = 0.481); therefore, both methods were used and combined for analysis.

The cups were checked for eggs every 24 h for up to 6 days. If eggs were found, the filter paper was removed, and the eggs were counted twice under a stereoscopic microscope. Any eggs that remained on the side of the cup were counted twice under the naked eye with a magnifying glass. Females that laid eggs were transferred to a new cup containing an oviposition substrate to provide the opportunity to lay additional eggs. After counting, eggs were washed in distilled water and transferred to small bowls or returned to the original cup, and thereafter kept in an incubator maintained at 27 °C. Larvae were checked daily and counted twice using pipettes. Eggs and larvae were recounted a third time if there was a difference greater than 5% between the two counts.

### 2.5. Dissection Procedure

Female mosquitoes were dissected 72 h after insecticide exposure and 76–88 h after blood-feeding (Figure 1). Preliminary tests were conducted to determine the timing of dissection in relation to the bloodmeal: it was observed that the stages of egg development were more distinctive at 72 h post-exposure than at 48 h between PPF-exposed and unexposed females. Therefore, a three-day minimum interval between blood-feeding and dissection was deemed optimal to allow sufficient time for bloodmeal assimilation and egg development. For females in the oviposition group, survivors were dissected 144 h post-exposure to allow sufficient time to lay eggs.

Prior to dissection, females were killed using ethyl acetate exposure for 10 min. Shortly after mosquito death, each female was placed on a glass slide along with a drop of 1X phosphate-buffered saline (PBS) solution under a stereoscopic microscope. Two needles were used to make an incision between the sixth and seventh tarsal segments of the abdomen to extract the eggs into PBS solution. Each female was scored based on the stage of egg development (Christophers’ stages) from I to V [15]. The number of eggs per female were counted twice, and a recount was performed if there was at least a 5% disparity between the two counts. The absence of eggs was also recorded, as some females were blood-fed but did not produce eggs.

After insecticide exposure, females were sorted randomly into individual cups with unique labels placed at the bottom of the cup. The investigator was blinded during dissections to reduce bias when scoring fecundity outcomes between treatment groups.

### 2.6. Outcomes

The fecundity outcomes calculated in the oviposition and dissection groups are described in Table 1. Females classified as “fecund” are those which laid eggs in the oviposition group or were observed to carry mature (stage V) eggs when dissected at 72 h. Those classified as “sterile” did not oviposit in the oviposition group or carried underdeveloped eggs (stages I–IV) in the dissection group.

### 2.7. Data Analysis

Induced sterility, reduction in fecundity, and reduction in reproductive rate were calculated using the equation below. Induced sterility was calculated using the proportion (%) of fecund females, reduction in fecundity used mean mature eggs (n) per female, and reduction in reproductive rate used mean larvae (n) per female.
$$Inhibition=\left(1-\left(\frac{treatedgroup}{untreatedgroup}\right)\right)\times 100$$

To assess the sensitivity and specificity of the oviposition and dissection methods, females were grouped into two categories: sterile or fecund. Females were considered fecund if they laid eggs (oviposition group) or had stage V (mature/developed) eggs when dissected (dissection group). Females were considered sterile if they did not lay eggs (oviposition group) or had stages I–IV (underdeveloped) eggs when dissected (dissection group). Sensitivity (true ‘sterile’ rate), specificity (true ‘fecund’ rate), positive predictive value (PPV; rate of ‘sterile’ results that are true ‘sterile’), negative predictive value (NPV; rate of ‘fecund’ results that are true ‘fecund’) were calculated using the equations in Table 2.

For analysis of blinded dissections, a correct prediction (%) was calculated as the proportion of true positives (sensitivity) predicted correctly in each treatment group. For Untreated net and Olyset^®^ net, this was calculated as the number of female mosquitoes predicted to be unexposed or fecund over the total number of females dissected in that treatment group. For the treatment groups exposed to PPF Net, PPF Net 20W, or Olyset^®^ Duo 20W, this was calculated as the number of female mosquitoes predicted to be PPF-exposed or sterile over the total number of females dissected in that treatment group.

Statistical analyses were performed using STATA 14 (StataCorp, College Station, TX, USA). Comparisons of proportions were analyzed using logistic regression reporting the odds ratio (OR), and comparisons of count data were performed with Poisson regression, reporting the incidence rate ratio (IRR). Linear regression was used to compare means. Graphs were made using GraphPad Prism 9 (GraphPad Software, Boston, MA, USA). 

## 3. Results

### 3.1. Comparison of Oviposition and Dissection Methods

A total of 707 blood-fed *An. gambiae* RSP females were exposed to LLINs in cylinder assays. In both the dissection and oviposition groups, female survival at 72 h post-exposure was 69.8% (95% CI: 65.1–74.2; n = 388) in the untreated arm and 57.6% (95% CI: 51.9–63.1; n = 302) in the PPF arm. At 144 h post-exposure in the oviposition group, survival was 40.2% (95% CI: 34.3–46.5; n = 246) in the untreated arm and lower (22.2%, 95% CI: 16.7–28.8; n = 185) in the PPF arm. Female survival was significantly different between the two treatment arms at both 72 h (OR: 0.59, *p* = 0.001) and 144 h (OR: 0.42, *p* < 0.001) post-exposure.

A total of 134 blood-fed females were dissected at 72 h post-exposure, and 278 females were followed up for egg-laying in the oviposition group up to 144 h post-exposure. The dissection method classified a greater proportion of untreated females as fecund than the oviposition method (71.2% (95% CI: 58.0–81.5) and 17.0% (95% CI: 12.2–23.3), respectively), indicating that not all bloodmeals led to a batch of eggs, and that the cues for oviposition were suboptimal under laboratory conditions (Table 3). Induced sterility comparing the PPF arm to the untreated arm was 100.0% in the dissection group and 96.8% in the oviposition group. The fecundity (mean mature eggs per female) of the untreated females was higher in the dissection group than in the oviposition group (29.9 and 10.8 eggs, respectively) because all eggs were accounted for in the dissection group but not all females laid eggs in the oviposition group. Reduction in fecundity was higher in PPF-exposed mosquitoes in the dissection group (100.0%, (0/29.9)) compared to the oviposition group (86.1%, (1.5/10.8)); this must have been by chance as both oviposition and dissection groups were exposed to PPF the same way.

Eggs laid by females in the oviposition group were followed up to measure fertility outcomes. In the untreated group, females laid eggs an average of 3.5 days following insecticide exposure, with a range of 3 to 6 days. One female in the PPF group laid eggs at 4 days post-exposure. Ovipositing females produced a total of 1023 larvae in the control group but only 17 larvae in the PPF group. The fertility (egg hatch rate) was 50.0% (95% CI: 34.4–65.7) and 22.8% in the untreated and PPF groups, respectively. This results in a reduction in fertility induced by PPF of only 54%. The reproductive rate (mean larvae per female) was 33 (95% CI: 20–46) and 17 in the untreated and PPF groups, respectively. The reduction in reproductive rate was 48.5%. By contrast, the PPF-induced sterility was 96.8–100%, making a larger contribution than fertility. This is most likely because PPF-induced sterility is due to reductions in fecundity and not fertility.

To confirm the development stage of the retained eggs in the non-ovipositing females, all females in the oviposition group that survived up to six days post-exposure were dissected (n = 135). The majority of the untreated (68.4%, n = 95) and PPF-exposed females (82.5%, n = 40) had retained eggs. Most eggs belonging to the untreated females (78.5%, 95% CI: 66.5–87.0; n = 65) were mature, whilst 100.0% of PPF-exposed females (n = 32) had underdeveloped eggs.

### 3.2. Sensitivity & Specificity Analysis

The sensitivity, specificity, positive predictive value (PPV), and negative predictive value (NPV) of each method (Figure 2) was calculated based on the number of sterile or fecund females (Appendix A). The oviposition method correctly identified almost all sterilized females exposed to PPF (99.1% sensitivity) and most females classified as fecund with this method were truly unexposed to PPF (96.9% NPV). However, based on the exposure, this method underestimated the number of unexposed females that should be fecund (18.9% specificity), as these mosquitoes failed to produce eggs after blood-feeding, and overestimated the proportion of females classified as sterile (45.9% PPV). The dissection method had 100.0% sensitivity and NPV, as all PPF-exposed females with underdeveloped eggs were classified as sterile and all unexposed females with mature eggs as fecund. The dissection technique underestimated approximately half the true number of unexposed females (52.5% specificity) but had more than two-fold the specificity as the oviposition method. The dissection method overestimated the true proportion of sterile females but had a higher positive predictive value (58.7%) than the oviposition method.

### 3.3. Dissections after PPF Exposure in Cylinder Assays and Tunnel Tests

A total of 768 blood-fed Muleba-Kis mosquitoes were exposed in 3 min cylinder bioassays to untreated net, PPF net, Olyset^®^ net, Olyset^®^ Duo, and Olyset^®^ Duo 20W (Table 4). There were no survivors following Olyset^®^ Duo exposure in cylinder assays (n = 95). Mortality after 24 h in cylinder assays ranged between 5.2% (95% CI: 2.7–9.7) in the untreated group and 11.6% (95% CI: 7.8–17.1) in the PPF group. There was no evidence for delayed mortality between 24 h and 72 h of holding.

In overnight tunnel tests, a total of 730 unfed Muleba-Kis mosquitoes were exposed to the same treatments, including a PPF 20W Net. Of those exposed, 60.5% (n = 205), 67.3% (n = 196), 100.0% (n = 50), 23.2% (n = 164), and 29.6% (n = 115) had bloodmeals in tunnels containing untreated net, PPF net, PPF 20W, Olyset^®^ net, and Olyset^®^ Duo 20W, respectively. Mortality after 24 h ranged from 8.8% (95% CI: 2.8–24.4) in the Olyset^®^ Duo 20W group to 18.2% (95% CI: 9.3–32.6) in the PPF 20W group. There was no evidence for delayed mortality between 24 h and 72 h after the tunnel test.

The majority (81.0–96.4%) of dissected females from both types of bioassays presented with eggs, while the rest did not undergo oogenesis. The proportion of females with mature eggs was the highest in the untreated net and Olyset^®^ net groups with no significant difference in both cylinder assays (OR: 1.02, 95% CI: 0.45–2.33; *p* = 0.963) or tunnel tests (OR: 3.14, 95% CI: 0.38–26.12; *p* = 0.291). On the other hand, PPF net exposure resulted in a significantly lower proportion of females with mature eggs compared to the untreated group from cylinder assays (OR: 0.002, 95% CI: 0.0005–0.01; *p* < 0.001) and tunnel tests (OR: 0.01, 95% CI: 0.004–0.03; *p* < 0.001). In the Olyset^®^ Duo 20W group, the proportion of females with mature eggs was also significantly lower in cylinder assays (OR: 0.29, 95% CI: 0.13–0.61; *p* = 0.001) and tunnel tests (OR: 0.07, 95% CI: 0.02–0.21; *p* < 0.001). No females exposed to PPF 20W in tunnel tests developed mature eggs.

Induced sterility (a proxy term for oviposition inhibition) was 97.9% in females exposed to PPF Net and 20.4% in those exposed to Olyset^®^ Duo 20W in cylinder assays. In tunnel tests, induced sterility was 91.6%, 100.0%, and 60.7% in females exposed to PPF net, PPF 20W, and Olyset^®^ Duo 20W, respectively. There was no induced sterility in females exposed to Olyset^®^ net in either cylinder assays or tunnel tests.

In cylinder assays, the mean number of mature eggs did not differ between the untreated and Olyset^®^ net groups (IRR: 1.02, *p* = 0.027; 95% CI: 1.0–1.03) but differed significantly between other treatment groups in reference to the control group: untreated net vs. PPF net (IRR: 0.44, *p* < 0.001; 95% CI: 0.35–0.56) and untreated net vs. Olyset^®^ Duo 20W (IRR: 1.04, *p* < 0.001; 95% CI: 1.03–1.05). In tunnel tests, the mean number of mature eggs was significantly different across all treatment groups in reference to the control group: untreated net vs. Olyset^®^ net (IRR: 1.05, *p* < 0.001; 95% CI: 1.02–1.08), untreated net vs. PPF net (IRR: 1.22, *p* < 0.001; 95% CI: 1.13–1.32), and untreated net vs. Olyset^®^ Duo 20W (IRR: 0.96, *p* = 0.002; 95% CI: 0.93–0.99).

Fecundity (mean mature eggs per female) from cylinder assays was 57.2, 0.6, 64.0, and 55.9 for the untreated, PPF net, Olyset^®^ net and Olyset^®^ Duo 20W groups, respectively. Reduction in fecundity was 99.0% in females exposed to PPF net and 2.3% for those exposed to Olyset^®^ Duo 20W in cylinders. In tunnel tests, fecundity was 59.9, 7.1, 0.0, 82.6, and 21.6 in the untreated net, PPF net, PPF 20W, Olyset^®^ net and Olyset^®^ Duo 20W groups, respectively. Reduction in fecundity was 88.1%, 100.0%, and 63.9% in females exposed to PPF net, PPF 20W, and Olyset^®^ Duo 20W, respectively. There was no reduction in fecundity for females exposed to Olyset^®^ net from either bioassay.

### 3.4. Blinded Dissections

To test the applicability of ovary dissection in practice, gravid females were dissected and a prediction of PPF exposure status (i.e., exposed to PPF or unexposed) was made by a blinded investigator based on the stages of egg development. Females with only mature eggs were predicted to be fecund or unexposed, and those with underdeveloped eggs were predicted to be sterile or PPF-exposed (Appendix A). Females containing eggs with both mature and underdeveloped eggs (mixed stages) were excluded from the predictions.

Based on the stage of egg development, 93.8%, 97.1%, 93.6%, and 18.9% of females exposed in cylinders to untreated net, PPF net, Olyset^®^ net and Olyset^®^ Duo 20W, respectively, had the correct exposure status predicted (Figure 3). In tunnel tests, correct predictions were 91.9%, 90.1%, and 95.8% for females exposed to untreated net, PPF net and Olyset^®^ net, respectively. All dissected females exposed to PPF 20W and Olyset^®^ Duo 20W were predicted correctly.

In cylinder assays, 2.6%, 1.9%, 1.6%, and 11.9% of those exposed to untreated net, PPF net, Olyset^®^ net and Olyset^®^ Duo 20W, respectively, had mixed stages of eggs and were therefore not given a prediction. In tunnel tests, 1.3%, 3.2%, and 14.8% of females had mixed stages of eggs when exposed to untreated net, PPF net and Olyset^®^ Duo 20W, respectively. There were no females with mixed stages when exposed to PPF 20W or Olyset^®^ net in tunnels.

## 4. Discussion

In this paper, upon comparing dissection and oviposition to assess the efficacy of PPF on fecundity outcomes, dissection was an effective technique for observing sterility in *An. gambiae* s.s. exposed to PPF. Two key fecundity outcomes (induced sterility and reduction in fecundity) were measured using oviposition and dissection, which produced similar results due to the strong sterilizing impact of PPF and the high sensitivities (oviposition: 99.1%; dissection: 100.0%) of both techniques. However, the proportion of fecund females in the untreated group was significantly higher when measured by dissection versus oviposition (71.2% vs. 17.0%), owing to the low oviposition rates of the mosquito colony. It was therefore more effective to directly measure fecundity by dissection.

Both techniques showed low specificity (oviposition: 18.9%; dissection: 52.5%). This is in part due to some untreated females being falsely classified as sterile (i.e., false positives) because they had underdeveloped eggs when dissected (oviposition: 31.6%; dissection: 28.4%). Possible reasons for this phenomenon are discussed in the Appendix (Appendix B). In the oviposition group, only a small proportion of untreated females laid eggs, despite the majority (68.4%) presenting with mature eggs when dissected six days post-exposure. Thus, dissection showed greater specificity, owing again to the low oviposition rates of the colony.

Next, we dissected *An. gambiae* females exposed to dual-AI nets impregnated with PPF and a pyrethroid. In cylinder assays, mosquitoes were exposed to net pieces for 3 min, and in tunnel tests, the exposure times were ad libitum. The holed net pieces create a natural barrier that host-seeking mosquitoes must cross to feed on a host (i.e., guinea pig), providing more realistic exposure to the AIs as they would encounter in real-life settings. In both assays, the PPF unwashed and 20W groups had the highest reductions in fecundity compared to all other treatment groups, as most eggs from PPF-exposed females arrested at an early stage of development (typically, Christophers’ stage III). In the Olyset^®^ Duo 20W group in both assays, the proportion of fecund females was higher than the PPF-exposed groups, but significantly lower than the untreated group, suggesting that there was still enough PPF to sterilize some mosquitoes, but not all. The proportion of fecund females did not differ between the untreated and Olyset^®^ net groups, indicating that permethrin had no impact on egg development. Similarly, another study found that eggs were 100% abnormal in *An. gambiae* females exposed to PPF net and 60% abnormal for those exposed to Olyset^®^ Duo, and they observed 0% abnormalities in those exposed to untreated net or Olyset^®^ net [14]. Moreover, the dissections were performed in this study by a blinded investigator to reduce bias when scoring sterility between treatment groups and test if predictions of prior PPF exposure could be made accurately. Correct predictions ranged from 90 to 100% across all treatment groups, confirming the high sensitivity of dissection.

PPF inhibits the action of juvenile hormone secreted by the corpora allata after a bloodmeal [2,16,17,18,19]. Eggs that would have normally reached follicular maturation are arrested at an early stage of development, resulting in underdeveloped eggs and sterility. PPF is dose-dependent and observed to be most effective in *An. gambiae* mosquitoes when encountered within a 24 h period before or after a bloodmeal [2,6,9]. Regardless of whether the female is parous or nulliparous, after a single exposure to PPF, oviposition inhibition may be lifelong [2,6,8]. In this study, the stages of egg development were clear between PPF-exposed (Christophers’ stage III) and unexposed (stage V) females dissected 72 h post-exposure (see Appendix A). Thus, dissecting gravid females is a feasible and practical technique to measure sterility.

In comparison to measuring oviposition rates, dissection would be ideal to measure PPF-induced sterility in wild mosquitoes exposed to PPF LLINs in field or semi-field studies, such as phase II experimental hut trials or phase III cluster randomized controlled trials (RCTs) [5]. Dissection is more operationally feasible and timely, as sterility can be rapidly assessed by observing the stages of egg development and it bypasses the need for wild mosquitoes to oviposit in the laboratory. The artificial conditions of the laboratory, the removal of the need to fly to search for breeding sites, and insufficient oviposition stimuli may inhibit the mosquito’s natural urge to oviposit [20]. In some studies, *An. gambiae* colonies adapted to insectary conditions produced oviposition rates as low as 25% (Kisumu strain), 34% (VKPER strain), 49% (RSP strain), and 0–100% (Dieldrin Dl^r-s^ strain) [2,6,7,21]. Overall, dissection as a substitute for oviposition would circumvent the need for forced oviposition under laboratory conditions and shorten the follow-up time to when females are expected to have fully developed eggs, thus minimizing the time, resources, and labor that are required.

## 5. Conclusions

This study demonstrated that clear morphological differences could be observed between the eggs of PPF-exposed and unexposed or permethrin-exposed blood-fed *An. gambiae* s.s. when dissected 3 days post-exposure. During blinded dissections, the correct exposure group was predicted with >90% accuracy. We hereby demonstrated that ovary dissection is a sensitive tool to measure sterility in *An. gambiae* females exposed to PPF-treated nets.

## Figures and Tables

**Figure 1 insects-14-00552-f001:**
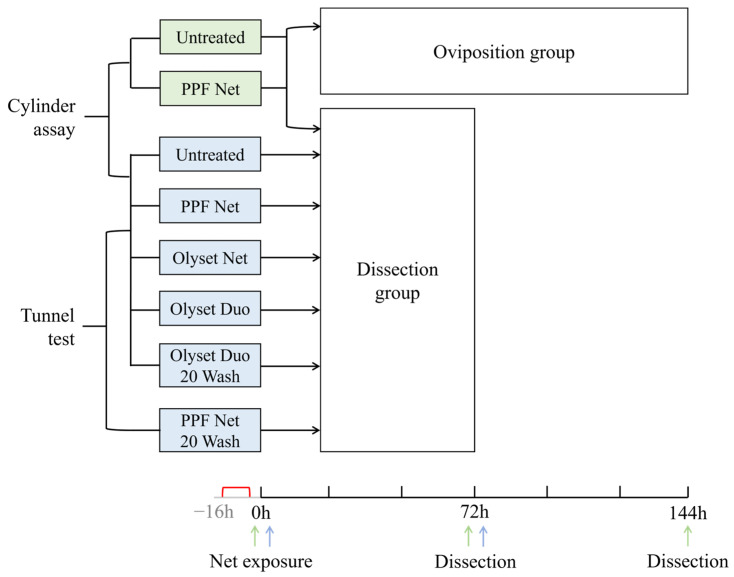
Schematic of bioassays, treatment groups, and timeline of mosquito blood-feeding, net exposure and dissection. The green treatments and arrows refer to the first experiment comparing the oviposition and dissection methods; the blue treatments and arrows refer to the remaining experiments comparing the bioassays and treatment groups. The red bracket indicates the period of blood-feeding prior to net exposure at 0 h. Mosquitoes in the dissection and oviposition groups were dissected at 72 h and 144 h post-exposure, respectively.

**Figure 2 insects-14-00552-f002:**
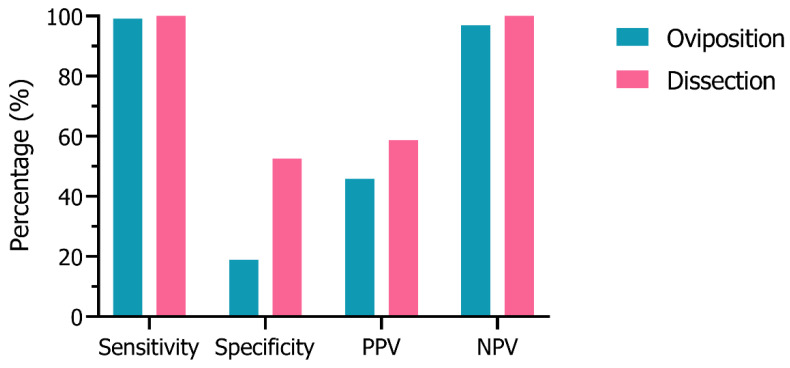
The percentage (%) sensitivity, specificity, positive predictive value (PPV), and negative predictive value (NPV) of the oviposition and dissection methods.

**Figure 3 insects-14-00552-f003:**
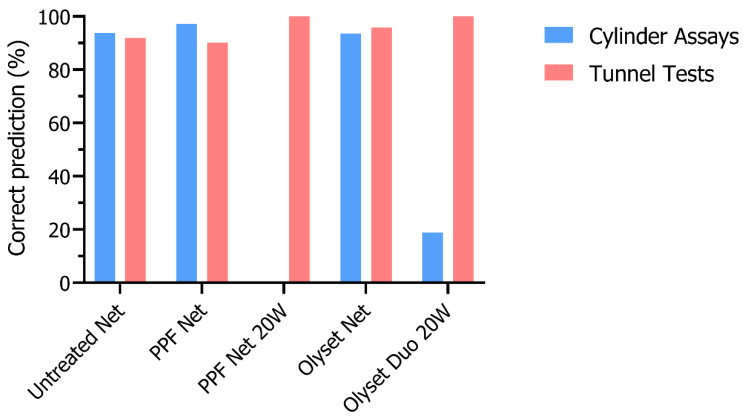
Correct prediction (%) of PPF exposure to *An. gambiae* females during blinded dissections in cylinder assays and tunnel tests.

**Table 1 insects-14-00552-t001:** Reproductive outcomes described for the oviposition and dissection methods. Outcomes are expressed as a mean number or as a proportion (%).

Outcome	Oviposition Group	Dissection Group
% Fecund females	Proportion of females that laid eggs over the total number of blood-fed females alive at 3 days post-exposure.	Proportion of dissected females with mature eggs over the total number of blood-fed females dissected at 3 days post-exposure.
% Induced sterility (proxy for oviposition inhibition)	Reduction in the proportion of egg-laying females caused by PPF exposure.	Reduction in the proportion of dissected females with mature eggs caused by PPF exposure.
Fecundity	Mean number of eggs laid per female, calculated as the total number of eggs laid divided by the total number of blood-fed females.	Mean number of mature eggs per female, calculated as the total number of mature eggs divided by the total number of blood-fed females.
% Reduction in fecundity	Reduction in the proportion of eggs laid per female caused by PPF exposure.	Reduction in the proportion of mature eggs dissected per female caused by PPF exposure.
Fertility	Proportion of laid eggs that hatched into larvae.	-
Reproductive rate	Mean number of larvae that hatched per blood-fed female alive at 3 days post-exposure.	-
% Reduction in reproductive rate	Reduction in the proportion of larvae that hatched per female caused by PPF exposure.	-

**Table 2 insects-14-00552-t002:** Calculations for the sensitivity, specificity, positive predictive value (PPV), and negative predictive value (NPV) of each method based on PPF exposure status.

	PPF Group	Untreated Group	Calculation
Sterile	A	B	PPV = A/(A + B)
Fecund	C	D	NPV = D/(C + D)
Calculation	Sensitivity = A/(A + C)	Specificity = D/(B + D)	

**Table 3 insects-14-00552-t003:** Fecundity and fertility results of *An. gambiae* RSP females in the oviposition and dissection groups.

	Oviposition		Dissection	
	Untreated	PPF	Untreated	PPF
N Survivors at 3 days post-exposure	164	114	80	54
N Survivors at 6 days post-exposure	99	41	-	-
N Dissected females with eggs	-	-	59	36
N Fecund females	31	1	42	0
% Fecund females (95% CI)	17.0 (12.2–23.3)	0.7 (0.1–4.9)	71.2 (58.0–81.5)	0.0
% Induced sterility	-	96.8	-	100.0
N Underdeveloped (stages II–IV) eggs	0 (not laid)	0 (not laid)	918	3510
N Developed (stage V) eggs	1778	75	2393	0.0
N Mean eggs per female	10.8	1.5	41.4	65.0
Fecundity: N Mean mature eggs per female	10.8	1.5	29.9	0.0
% Reduction in fecundity	-	86.1	-	100.0

**Table 4 insects-14-00552-t004:** Mortality and reproductive outcomes *An. gambiae* s.s. Muleba-Kis females dissected 3 days post-exposure to untreated net, PPF net, PPF 20W, Olyset^®^ net, and Olyset^®^ Duo 20W in cylinder assays and/or tunnel tests.

	Cylinder Assay				Tunnel Test				
	Untreated	PPF	Olyset^®^ Net	Olyset^®^ Duo 20W	Untreated	PPF	PPF 20W	Olyset^®^ Net	Olyset^®^ Duo 20W
N Exposed	174	189	181	129	205	196	50	164	115
N Blood-fed (tunnel tests)	-	-	-	-	124	132	50	38	34
% Mortality 24 h (95% CI)	5.2 (2.7–9.7)	11.6 (7.8–17.1)	9.4 (5.9–14.6)	9.3 (5.3–15.7)	13.7 (8.7–21.0)	9.4 (5.4–15.8)	18.2 (9.3–32.6)	8.6 (2.7–23.8)	8.8 (2.8–24.4)
N Females dissected	142	113	145	93	90	98	28	25	30
% Females with eggs, all stages (95% CI)	81.0 (73.6–86.7)	93.8 (87.5–97.0)	89.0 (82.7–93.1)	90.3 (82.4–94.9)	83.3 (74.1–89.7)	95.9 (89.5–98.5)	96.4 (77.8–99.5)	96.0 (75.5–99.5)	86.7 (68.9–95.0)
N Females with mature eggs	105	2	116	60	66	7	0	23	9
% Females with mature eggs (95% CI)	89.7 (82.7–94.1)	1.9 * (0.5–7.3)	89.9 (83.4–94.1)	71.4 * (60.8–80.1)	88.0 (78.4–93.7)	7.4 * (3.6–14.9)	0.0 *	95.8 (74.7–99.4)	34.6 * (18.8–54.8)
% Induced sterility	-	97.9	0.0	20.4	-	91.6	100.0	0.0	60.7
N Eggs, all stages (95% CI)	8422 (7658–9186)	8399 (7965–8832)	9705 (8897–10,512)	6598 (5848–7347)	5926 (5347–6505)	7582 (7094–8070)	1974 (1764–2183)	2176 (1928–2424)	1590 (1310–1869)
N Mature eggs dissected (n, 95% CI)	8127 (7391–8863)	68 (0–159)	9285 (8511–10,059)	5202 (4542–5862)	5392 (4828–5956)	699 (481–916)	0	2065 (1818–2311)	648 (421–875)
N Mean eggs dissected, all stages (SD)	75.2 (36.7)	81.5 (21.7)	77.6 (36.8)	78.5 (41.6)	79.0 (34.0)	80.7 (25.5)	73.1 (20.5)	90.7 (25.7)	61.1 (27.8)
N Mean mature eggs dissected (SD)	77.4 (36.5)	34.0 * (32.5)	80.0 (36.5)	86.7 * (43.3)	81.7 (35.0)	99.8 * (41.5)	0.0	89.8 * (25.9)	72.0 * (38.2)
Fecundity: N Mean mature eggs per female	57.2	0.6	64.0	55.9	59.9	7.1	0.0	82.6	21.6
% Reduction in fecundity	-	99.0	0.0	2.3	-	88.1	100.0	0.0	63.9

* Significantly different results compared to untreated net group by logistic regression or linear regression (*p* < 0.05).

## Data Availability

The datasets generated and/or analyzed during the current study are available from the corresponding author upon reasonable request.

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
