# Peer review of "Ovary Dissection Is a Sensitive Measure of Sterility in Anopheles gambiae Exposed to the Insect Growth Regulator Pyriproxyfen"

_insects, 2023, doi:10.3390/insects14060552_

Round 1

Reviewer 1 Report

This study used the observation of oviposition and ovarial dissection to evaluate the effectiveness of Pyriproxyfen (PPF) in sterilizing female mosquitoes when used in combination with insecticidal nets, and investigated whether dissecting the ovaries could serve as a reliable proxy for evaluating sterility in Anopheles gambiae mosquitoes. The results suggest that dissecting the ovaries is a sensitive technique for assessing sterility in female Anopheles gambiae mosquitoes and can serve as a predictor of PPF exposure. The experiments were well designed and performed, providing a specific guideline for evaluating the bio-efficacy of LLINs containing growth regulators. However, there are some questions to be addressed before it can be accepted for publication.

It would be beneficial to include an introduction explaining the lifecycle of Anopheles mosquitoes and their role in transmitting malaria. This information would help readers understand the significance of the study.

It is recommended to include a diagram illustrating the bioassay procedures, including the different treatments used. This visual aid would enhance the understanding of the experimental setup.

Please provide details on how the blood-feeding was conducted using a live guinea pig. Did you use any drugs to anesthetize the pig during mosquito blood feeding?

How were the mosquitoes maintained after blood feeding and PPF treatment? Were they provided with water or sugar water? It is important to consider whether the sugar or water provided could affect the assays.

Without an oviposition substrate, mosquitoes with eggs may break the eggs and reuse the nutrients, potentially impacting the oviposition assay. Additionally, choosing more than six days post blood meal for the oviposition assay raises concerns about the potential effects of prolonged egg arrest on the results. This may also explain why some untreated females had underdeveloped eggs.

Please specify the duration of the ethyl acetate treatment used to kill the mosquitoes before dissection. Also, consider whether the use of ethyl acetate could affect egg development. Exploring alternative methods such as anesthetizing the mosquitoes in a cold room before dissection could be worth considering.

It would be helpful to provide information on how the LLINs obtained from Sumitomo Chemical Co. Ltd were washed. 

Author Response

Thank you very much for taking the time to review the manuscript and for providing helpful comments and suggestions. Please find below our responses to your points.

Point 1: It would be beneficial to include an introduction explaining the lifecycle of Anopheles mosquitoes and their role in transmitting malaria. This information would help readers understand the significance of the study.

Response to point 1: We agree, so a brief introduction including malaria transmission and the reproductive cycle of Anopheles mosquitoes was included (lines 48-52).

Point 2: It is recommended to include a diagram illustrating the bioassay procedures, including the different treatments used. This visual aid would enhance the understanding of the experimental setup.

Response to point 2: You are correct, and so we added a schematic including the bioassays, treatments, and timing of the experiments. We replaced Fig. 1 (study timeline) with this new schematic and updated the legend accordingly (lines 153-160).

Point 3: Please provide details on how the blood-feeding was conducted using a live guinea pig. Did you use any drugs to anesthetize the pig during mosquito blood feeding?

Response to point 3: We expanded on how guinea pigs were sedated (using xylazine) and placed over a mesh cage to allow mosquitoes to feed through (lines 96-98).

Point 4: How were the mosquitoes maintained after blood feeding and PPF treatment? Were they provided with water or sugar water? It is important to consider whether the sugar or water provided could affect the assays.

Response to point 4: The mosquitoes were maintained on 10% glucose solution as stated in lines 119 and 129. We added the method of glucose delivery (on cotton pads) to these lines for added clarification. All mosquitoes tested in this study were maintained on sugar the same way, therefore there was probably no effect of the sugar on the assays.

Point 5: Without an oviposition substrate, mosquitoes with eggs may break the eggs and reuse the nutrients, potentially impacting the oviposition assay. Additionally, choosing more than six days post blood meal for the oviposition assay raises concerns about the potential effects of prolonged egg arrest on the results. This may also explain why some untreated females had underdeveloped eggs.

Response to point 5: Females with underdeveloped eggs were observed in all treatment groups and for both the oviposition and dissection methods. In the oviposition group, females laid eggs an average of 3.5 days after feeding and insecticide exposure, with a range of 3 to 6 days, showing they are still fecund up until at least 6 days. Therefore, having undeveloped eggs does not appear to be affected by oviposition assay or time until dissection.

Point 6: Please specify the duration of the ethyl acetate treatment used to kill the mosquitoes before dissection. Also, consider whether the use of ethyl acetate could affect egg development. Exploring alternative methods such as anesthetizing the mosquitoes in a cold room before dissection could be worth considering.

Response to point 6: A clarification was added to lines 169-170 to include the length of ethyl acetate exposure (10 min) and that immediately after exposure the mosquitoes were dissected. A previous study showed that ethyl acetate does not impact insect fecundity: https://www.tandfonline.com/doi/pdf/10.1080/00379271.2010.10697677

Point 7: It would be helpful to provide information on how the LLINs obtained from Sumitomo Chemical Co. Ltd were washed.

Response to point 7: In lines 111-112, it is written that the nets were washed following standard WHO guidelines, with a reference to their guidelines which are freely available online. It would take up quite some space to write the full washing procedure, and since it is not a key focus of the manuscript we argue to leave this statement as it is.

Reviewer 2 Report

Dear authors,

This is a very interesting research article that assesses ovary dissection method over oviposition method as a measure of sterility in Anopheles gambiae mosquitoes exposed to Pyriproxyfen based long-lasting nets.

Herein, you will find some comments for clarification and manuscript revision.

Lines 36-37: it should be “Both techniques demonstrated high sensitivity (oviposition: 99.1%; dissection: 100.0%) in the PPF treated nets, but specificity was significantly higher in the dissection group (52.5% vs. 18.9%) in the untreated nets.

Lines 166-168: It is not necessary to have a separate paragraph for these lines, and better incorporate to the previous paragraph.

Lines 185-191: Please explain in the text the meaning of the terms sensitivity, specificity, PPV and NPV.

Lines 229-231: It is stated that the dissection method classified a greater proportion of untreated females as fecund than the oviposition method, indicating that not all bloodmeals led to a batch of egg and that the cues for oviposition were suboptimal under laboratory conditions. Why the laboratory conditions were not favorable for egg laying of An. gambiae, considering that this is a mosquito that normally can breed under laboratory conditions? This seems like bias against oviposition method, hence making troublesome and probably not reliable the comparison between the two methods (dissection Vs oviposition). Please clarify.

Lines 369-370: The text in Appendix I could be limited and better be presented in the main text in the discussion.

Author Response

Thank you very much for taking the time to review the manuscript and for providing helpful comments and suggestions. Please find below our responses to your points.

Point 1: Lines 36-37: it should be “Both techniques demonstrated high sensitivity (oviposition: 99.1%; dissection: 100.0%) in the PPF treated nets, but specificity was significantly higher in the dissection group (52.5% vs. 18.9%) in the untreated nets.

Response to point 1: Thank you for the correction; we included a clarification as you suggested in the abstract at lines 35-37.

Point 2: Lines 166-168: It is not necessary to have a separate paragraph for these lines, and better incorporate to the previous paragraph.

Response to point 2: You are right, so we have removed the paragraph header at Line 178 to incorporate the two paragraphs.

Point 3: Lines 185-191: Please explain in the text the meaning of the terms sensitivity, specificity, PPV and NPV.

Response to point 3: Thank you for noticing that these terms were never explained. On lines 203-205 a brief description was added to clarify the definitions of sensitivity, specificity, PPV and NPV.

Point 4: Lines 229-231: It is stated that the dissection method classified a greater proportion of untreated females as fecund than the oviposition method, indicating that not all bloodmeals led to a batch of egg and that the cues for oviposition were suboptimal under laboratory conditions. Why the laboratory conditions were not favorable for egg laying of An. gambiae, considering that this is a mosquito that normally can breed under laboratory conditions? This seems like bias against oviposition method, hence making troublesome and probably not reliable the comparison between the two methods (dissection Vs oviposition). Please clarify.

Response to point 4: It is indeed not ideal for laboratory colonies to produce low oviposition rates, but this is the key disadvantage of using oviposition to measure fecundity. I believe this point is well addressed in the discussion section (lines 422-430), and includes references to other papers that observed similar low oviposition rates with their lab colonies. This is also a reason why we argue dissection is simpler and more sensitive for measuring sterility in mosquitoes, because whether or not they lay eggs, dissection can measure directly the impact of PPF on egg development.

Point 5: Lines 369-370: The text in Appendix I could be limited and better be presented in the main text in the discussion.

Response to point 5: While this paragraph could indeed be shortened and presented in the discussion, we argue that it distracts from the key points of the paper, and therefore we would like to keep it separate from the main discussion.

Round 2

Reviewer 2 Report

Dear authors,

Thank you for considering my comments and for your reply. In my view, the revised version of the article is suitable for publication.